Dynamic changes in prescription opioids from 2006 to 2017 in Texas

Ighodaro Ebuwa O. 1
McCall Kenneth L. 2
Chung Daniel Y. 1
Nichols Stephanie D. 2 4
Piper Brian J. bpiper@som.geisinger.edu psy391@gmail.com 1 3
1 Department of Medical Education, Geisinger Commonwealth School of Medicine , Scranton , PA , United States of America
2 Department of Pharmacy Practice, University of New England , Portland , ME , United States of America
3 Center for Pharmacy Innovation and Outcomes , Forty Fort , PA , United States of America
4 Department of Psychiatry, Tufts University , Medford , MA , United States of America
Connor Mark
Electronic publication date: 2019 Dec 6
Publication date: 2019
Volume: 7
Electronic Location ID: e8108
Received 2019 Aug 20; Accepted 2019 Oct 28
Copyright: ©2019 Ighodaro et al.
Copyright year: 2019
Copyright holder: Ighodaro et al.
License: This is an open access article distributed under the terms of the Creative Commons Attribution License, which permits unrestricted use, distribution, reproduction and adaptation in any medium and for any purpose provided that it is properly attributed. For attribution, the original author(s), title, publication source (PeerJ) and either DOI or URL of the article must be cited.
License URL: https://creativecommons.org/licenses/by/4.0/

Keywords: Pain, Addiction, Epidemiology, Drug, Policy, Buprenorphine, Codeine, Fentanyl, Hydrocodone, Opiate substitution treatment

Funding: Fahs-Beck Fund for Research and Experimentation NIEHS T32-ES007060-31A1 Health Resources Services Administration D34HP31025 Brian J. Piper, Daniel Y. Chung and Stephanie D. Nichols were supported by the Fahs-Beck Fund for Research and Experimentation. Software was provided by NIEHS (T32-ES007060-31A1). Brian J. Piper is supported by Health Resources Services Administration (D34HP31025). The funders had no role in study design, data collection and analysis, decision to publish, or preparation of the manuscript.

==============================
Background

The US is experiencing an epidemic of opioid overdoses which may be at least partially due to an over-reliance on opioid analgesics in the treatment of chronic non-cancer pain and subsequent escalation to heroin or illicit fentanyl. As Texas was reported to be among the lowest in the US for opioid use and misuse, further examination of this state is warranted.

Materials and Methods

This study was conducted to quantify prescription opioid use in Texas. Data was obtained from the publicly available US Drug Enforcement Administration’s Automation of Reports and Consolidated Orders System (ARCOS) which monitors controlled substances transactions from manufacture to commercial distribution. Data for 2006–2017 from Texas for ten prescription opioids including eight primarily used to relieve pain (codeine, fentanyl, hydrocodone, hydromorphone, meperidine, morphine, oxycodone, oxymorphone) and two (buprenorphine and methadone) for the treatment of an Opioid Use Disorder (OUD) were examined.

Results

The change in morphine mg equivalent (MME) of all opioids (+23.3%) was only slightly greater than the state’s population gains (21.1%). Opioids used to treat an OUD showed pronounced gains (+90.8%) which were four-fold faster than population growth. Analysis of individual agents revealed pronounced elevations in codeine (+387.5%), hydromorphone (+106.7%), and oxycodone (+43.6%) and a reduction in meperidine (−80.3%) in 2017 relative to 2006. Methadone in 2017 accounted for a greater portion (39.5%) of the total MME than hydrocodone, oxycodone, morphine, hydromorphone, oxymorphone, and meperidine, combined. There were differences between urban and rural areas in the changes in hydrocodone and buprenorphine.

Conclusions

Collectively, these findings indicate that continued vigilance is needed in Texas to appropriately treat pain and an OUD while minimizing the potential for prescription opioid diversion and misuse. Texas may lead the US in a return to pre-opioid epidemic prescription levels.

Introduction

Rates of opioid use and misuse in Texas are historically low compared to other states and regions of the country. Approximately one out of every twenty (4.6%) Texas survey respondents reported using opioid analgesics in 2009 nonmedically (Centers for Disease Control and Prevention, 2011). Hydrocodone accounted for three-fifths of the calls involving opioids made to the Texas Poison Center Network between 2000 and 2010 (Forrester, 2012). Pill identification calls for hydrocodone in the Texas zip codes with large military bases increased from 2002 to 2009 by 463% (Ng et al., 2017). Self-reported past year heroin use in 2015 was one-third national levels (Substance Abuse and Mental Health Services Administration, 2017). Opioid overdoses are generally lower than other states with similar reporting quality; however, overdoses involving synthetic opioids other than methadone increased by 28.6% from 2015 to 2016 (Seth et al., 2018). Rates of Neonatal Abstinence syndrome in the West South Central Census region were the lowest in the US and less than one fifth of those of New England (Patrick et al., 2015). Texas ranked 46th in the US for per capita morphine mg equivalents (MME, 578) for ten prescription opioids and was approximately one-quarter the highest state (Rhode Island = 2,624) (Piper et al., 2018). Texas had a similar ranking (45th) for buprenorphine and methadone for Opioid Use Disorder (OUD) treatment  (Piper et al., 2018). Methadone use from Texas opioid treatment programs remained stable from 2011 to 2015 but buprenorphine quadrupled (Substance Abuse and Mental Health Services Administration, 2017). However, many counties, particularly in west and north Texas, lacked buprenorphine waivered physicians (Andrilla, Coulthard & Larson, 2017). Counties in eastern and northern Texas had greater per capita prescription opioid use (Alexander et al., 2019). Further, counties with more white residents had significant more opioid pills prescribed than more diverse counties (Alexander et al., 2019). Per capita calculations based on US Census data with the total population estimates should be interpreted with caution as undercounting, estimated at 239,500 in 2010 in Texas, may disproportionately impact minorities (Davis & Mulligan, 2019).

There are several policy changes and demographic characteristics which may contribute to the rates of prescription opioid use and misuse. The Texas Prescription Monitoring Program, operated by the Department of Public Safety until 2017 when oversight was transferred to the State Board of Pharmacy, began collecting information about Schedule II prescription drugs in July, 1982 and Schedule III–V drugs in September, 2008 (Prescription Drug Monitoring Program Training and Technical Assistance Center, Texas State Profile, 2019). Texas implemented legislation to combat “pill mills” in September, 2010 (Lyapustina et al., 2016). The October, 2014 federal reclassification of hydrocodone from Schedule III to Schedule II by the Drug Enforcement Administration (DEA) was another policy change to combat the opioid crisis. A study that compared exposures from six-months before to six-months after the heightened regulation found that this restriction was followed by a decrease in hydrocodone exposures but also increases in codeine, oxycodone and tramadol reports to Texas Poison Centers  (Haynes et al., 2016). Federal policies including the Comprehenisive Addiction and Recovery Act and 21st Century Cares Act in 2016 may have increased capacity to deliver OUD treatments.

Texas had the highest percent (9.4%) of uninsured children (<18) and uninsured adults (18–65, 19.0%) in 2015 in the US  (National Center for Health Statistics, 2017). The opioid epidemic has been described as iatrogenic (Piper et al., 2018; Wright et al., 2014). Economic disparities may, paradoxically, be protective at a population level against over-diagnosis or over-treatment with prescription opioids (Cabera et al., 2018). Persons with non-white ethnicity also had lower drug poison deaths involving opioid analgesics (Manchikanti et al., 2018). Perhaps the most important policy factor underlying prescription misuse is supply. The US Drug Enforcement Administration reduced opioid production quotas by ≥25% in 2017, particularly for hydrocodone (Manchikanti et al., 2018). There have been concerns that rural patients are less likely to be able to access buprenorphine prescribers (Andrilla et al., 2019). Therefore, as Texas has been reported to be among the lowest in the US in opioid use (Centers for Disease Control and Prevention, 2011) and misuse (Patrick et al., 2015; Piper et al., 2018), there may be some insights from this populous and diverse state which could be beneficial for others. The goal of this pharmacoepidemiological investigation was to identify and quantify any changes in prescription opioids over the past decade in Texas.

Materials & Methods

Procedures

Drug manufacturers and distributors report controlled substance transactions to the DEA as required by the 1970 Controlled Substances Act. This information is made publicly available by the Automated Reports and Consolidated Ordering System (ARCOS) (US Department of Justice, Drug Enforcement Administration, Division of Diversion Control, 2019). ARCOS includes prescription data for Veterans Affairs patients, military personnel receiving care at non-Veterans Affairs pharmacies, Indian Health Services, dispensing practitioners (e.g., veterinarians) and Narcotic treatment programs (NTP) that may not be included in other databases like Prescription (Drug) Monitoring Programs (PMP). NTPs in Texas reported to ARCOS increased by 15.6% from 2006 to 2017. Ten prescription opioids were selected based on prior research (Piper et al., 2018; Cabera et al., 2018; Feickert et al., 2019). Eight of these (hydrocodone, oxycodone, fentanyl, morphine, hydromorphone, oxymorphone, codeine and meperidine) are primarily used to relieve pain, and two (buprenorphine and methadone) are employed for an OUD. Although ARCOS is a national program and PMPs are operated at the state level, a high correspondence between ARCOS and a state PMP for oxycodone (r = 0.99) was recently reported (Piper et al., 2018). Population data (28.3 million; 42.0% non-Hispanic white in 2017) was obtained from the US Census. Procedures were approved by the Institutional Review Board of the University of New England (#20180410-009).

Data analysis

Three analyses were completed for this descriptive pharmacoepidemiological report. First, the MME for each opioid was determined. Conversions were completed with the following multipliers: buprenorphine 10, codeine 0.15, fentanyl 75, hydrocodone 1, hydromorphone 4, meperidine 0.1, morphine 1, oxycodone 1.5, and oxymorphone 3. Methadone from NTP had a conversion of 12 but 8 from other sources (Piper et al., 2018; Cabera et al., 2018; Feickert et al., 2019). Second, the weight in kg of each opioid was obtained for each year from 2006 to 2017. Percent change for each opioid, the two for an OUD, the nine primarily used for pain (including methadone from non-NTP sources), and all ten were calculated relative to 2006 and expressed as MME or kg of each agent. Third, heat maps of the percent change in the weight of hydrocodone or buprenorphine from 2012 to 2017 in each of the forty-nine three-digit zip codes reported by ARCOS were constructed with QGIS. Other figures were prepared with GraphPad Prism version 6.07.

Results

The Texas population increased by 4.94 million between 2006 and 2017. Figure 1A shows that the MME of all opioids peaked in 2013 but only slightly (+2.2%) exceeded the gains in population. Opioids for pain peaked in 2011 and subsequently returned to 2006 levels. In contrast, the agents employed to treat an OUD grew over four-fold faster than the population.

Figure 1 Percent change in weight, relative to 2006, of opioids used to treat pain (hydrocodone = 3,064.0, oxycodone = 922.2, codeine = 822.1, morphine = 765.8, methadone = 278.2, meperidine = 235.8, hydromorphone = 46.2, fentanyl = 22.5, oxymorphone = 3.3 kg), an opioid use disorder (OUD, methadone = 249.6, buprenorphine = 12.9 kg), or all (pain + OUD) by Morphine Mg Equivalent (A) or by weight (B) in Texas as reported to the Drug Enforcement Administration.

Percent change versus 2006 is shown in parentheses.

Figure 1B depicts the changes in individual agents. Codeine showed a pronounced elevation from 2014 until 2017 and hydrocodone exhibited a reduction during this period. Hydromorphone doubled from 2006 to 2017 while the increases in oxycodone (+46.6%) was double the population change (+21.1%). In contrast, meperidine showed a protracted decrease. Oxymorphone grew from 3.3 to 73.0 kg (+2,091.6%) in 2011, decreased slightly to 64.8 kg (+1,845.6%) in 2016, and to only 46.2 kg (+1,286.4%) in 2017. Methadone administered from non-NTPs (i.e., primarily pharmacies) was 278.2 kg in 2006, peaked at 293.5 kg in 2010, and declined to 153.2 kg in 2017. Methadone from NTPs grew by 64.0% from 249.6 kg in 2006 to 408.8 kg in 2017 (i.e., three-fold more quickly than population). Buprenorphine expanded, almost logarithmically, from 12.9 kg in 2006 to 105.3 kg in 2017 (Supplemental Information 1).

Figure 2 shows the percent of the total MME for each of ten-opioids in 2017. Methadone from NTPs was the top opioid and accounted for almost one-third of the total MME. Hydrocodone, oxycodone, and fentanyl combined were responsible for two-fifths (41.1%) of the MME.

Figure 2 Morphine mg equivalents of ten opioids in Texas in 2017 as reported to the Drug Enforcement Administration’s Automation of Reports and Consolidated Orders System.

Opioid Use Disorder: OUD.

Figure 3A shows the percent decrease in hydrocodone, by weight, in the last five years. With the exception of Amarillo and rural areas outside Lubbock and Austin, pronounced (≥32.5%) decreases were observed throughout the state. Figure 3B depicts that the percent increase in buprenorphine over the last half-decade was distributed across Texas.

Figure 3 Heat maps showing the percent decreases in hydrocodone (A) and increases in buprenorphine (B) from 2012 to 2017 in Texas.

Heat map showing the percent decreases in hydrocodone from 2012 to 2017 in Texas.

Discussion

There were dynamic, pronounced and agent specific changes in prescription opioids over the past decade in Texas. These results may be informative for other states or countries. The MME of all ten opioids peaked in 2013 and has been subsequently declining. Collectively, drugs used for the treatment of pain reached a maximum in 2011 and decreased each year thereafter to return to 2006 levels. Opioids for an OUD increased over ninety-percent. Nationwide, prior findings conducted either by the CDC (Guy et al., 2017) or using ARCOS (Piper et al., 2018) have found that prescription opioids for pain peaked around 2011 and have subsequently been declining (Raji et al., 2018). The rapid population growth in Texas may have delayed the temporal inflection point when more cautious opioid use for chronic non-cancer pain became evident. These findings can be contrasted with other ARCOS opioid reports from Arkansas (Sahota & Boyle, 2019), Colorado (Kropp et al., 2019), Delaware (Feickert et al., 2019; Davis et al., in press), Florida (Cabera et al., 2018), Hawaii (Cabera et al., 2018; Davis et al., in press), Indiana (Feickert et al., 2019; Davis et al., in press), Kentucky (Feickert et al., 2019; Davis et al., in press), Louisiana (Sahota & Boyle, 2019), Maine (Collins et al., 2019; Simpson et al., 2019), Maryland (Davis et al., in press; Kropp et al., 2019), Michigan (Feickert et al., 2019; Kropp et al., 2019), Minnesota  (Davis et al., in press), Nevada (Davis et al., in press), New Jersey (Feickert et al., 2019; Davis et al., in press), New Mexico (Sahota & Boyle, 2019), New York (Davis et al., in press), Ohio (Feickert et al., 2019), Oklahoma (Sahota & Boyle, 2019), Pennsylvania (Collins et al., 2019; Feickert et al., 2019), Utah  (Kropp et al., 2019), Vermont (Davis et al., in press), Washington (Wang et al., 2018), West Virginia (Feickert et al., 2019; Hatcher, Vaddadi & Nafziger, 2019), and the US Territories (Cabera et al., 2018).

Texas showed some parallels with the rest of the US when examining individual agents but also some differences. Meperidine showed a precipitous decline nationally (Piper et al., 2018; Collins et al., 2019) and in Texas over the last decade. This is possibly due to increasing concerns relating to CNS excitability and seizure potential (Yaksh & Wallace, 2017) and the Libby Zion case (Lerner, 2009). Concerns about its adverse reactions and potentially fatal drug interactions has prompted many to recommend restriction of its use or removal from the health-care system (Benner & Durham, 2011). Similarly, buprenorphine has rapidly increased in Texas and the US (Piper et al., 2018; Collins et al., 2019) possibly due to safety concerns about use of methadone in NTPs (Maxwell, Pullum & Tannert, 2005) and to increased access outside of NTPs, particularly in rural areas (Andrilla et al., 2019). Interestingly, the volume of codeine has been stable nationwide (Piper et al., 2018) but showed a striking elevation in Texas. The rescheduling of hydrocodone had a clear impact (i.e., a 13.2% reduction occurring relative to a 21.1%% increase in population) and appears to be offset by a transition to other agents like codeine. These findings are congruent with those of others (Bernhardt et al., 2017). Percent increases and later reductions for oxymorphone should be interpreted within the context that the Opana ER formulation was only FDA approved for moderate to severe pain in 2006. The nationwide pattern is of a decrease in oxymorphone since 2011, likely resulting from the HIV cases in Indiana involving oxymorphone misuse  (Peters et al., 2016) and the voluntary withdrawal of Opana ER. There are pronounced regional differences in use of the strong opioids hydrocodone (Piper et al., 2018) and oxycodone. Hydrocodone was prescribed six times more commonly than oxycodone in Indiana (Wright et al., 2014). Oxycodone, by MMEs, was employed almost three-fold more than hydrocodone nationally (Piper et al., 2018).

Texas ranked 45th in the US for per capita use of prescription fentanyl  (Collins et al., 2019). Nationally, fentanyl distribution decreased by eighteen percent between 2016 and 2017 (Collins et al., 2019). In contrast, fentanyl in Texas has been relatively stable. Further monitoring of prescription fentanyl use versus nonmedical use of pharmaceutical and illicitly manufactured fentanyl should continue to be a priority.

There is some evidence that rural areas may have greater opioid misuse  (Keyes et al., 2014). The characteristics of rural areas that contribute to this profile is older age (Piper et al., 2018) and a greater white population (Alexander et al., 2019). Prescription opioid use decreased more after 2017 in Texas counties with more (>80%) whites (Alexander et al., 2019). It would be challenging to view the observed reductions in hydrocodone which were largest in El Paso (population = 679 K), Brownsville (183 K), Houston (2.31 million) and Jasper (7.6K) as solely being driven by population size. Some of the declines may reflect regression to the mean and a return to more temperate prescribing practices in areas that started at the highest levels (Alexander et al., 2019).

Texas has the unfortunate distinction of leading the country for the highest percent of the population lacking health insurance (17%) (Kaiser Family Foundation, 2017). This is almost twice the national level (9%) and over four-fold higher than West Virginia (4%) or Massachusetts (3%) (Kaiser Family Foundation, 2017). The opioid epidemic has been described as iatrogenic (Piper et al., 2018; Wright et al., 2014). Prescription opioids were the first opioid among two-thirds of substance abuse treatment program admissions with heroin dependence that initiated opioid abuse in the 2010s. This is very different than the 1960s when less than one-fifth of opioid “abusers” (currently designated as OUD) started on prescription opioids (Cicero et al., 2014). Possibly, economic factors could act as disincentives to limit chronic opioid use and decrease the likelihood of escalation to heroin or illicit fentanyl. Conversely, an absence of health insurance may contribute to an underutilization of evidence-based pharmacotherapies for an OUD like methadone (Cicero et al., 2014) or buprenorphine (Mattick et al., 2014) or to increased self-medication for pain, depression, anxiety, or sleep problems with non-prescribed opioids. Further national policy research (Davis et al., in press; Collins et al., 2019) is necessary to examine how variations in health insurance, including Medicaid expansion, may be associated with pharmacoepidemiological differences in the long-term treatment of pain or opioid misuse.

There are some strengths and limitations to this report and caveats of this data. ARCOS is a comprehensive data source in terms of including diverse patient groups often omitted from other studies and is publicly available (US Department of Justice, Drug Enforcement Administration, Division of Diversion Control, 2019). For example, the Texas Prescription Monitoring Program is prohibited from reporting methadone or buprenorphine administered from NTPs. This is unfortunate because methadone from NTPs accounts for the most MMEs of any opioid in Texas and nationally (Piper et al., 2018). Persons with an OUD are currently denied the benefits of the Texas PMP due to 42 Code of Federal Regulations Part 2 in that their health care providers, outside of the narcotic treatment program, are prevented from being able to know about their OUD opioid pharmacotherapies. Unlike ARCOS, the Texas PMP may not consistently report on opioids from the Veterans Affairs, Indian Health Services, veterinarians, dentists, or Emergency Rooms. However, ARCOS does not report on all opioids that may be of interest (e.g., tramadol). The population information in Fig. 1 is only as accurate as that reported by the US Census and is likely an under-estimate due to under-counting of undocumented persons. The illegal immigrant population nationwide was estimated to be 12.2 million in 2007 and decreased to 11.3 million in 2016 (Gomez, 2019). An accurate 2020 Census will continue to be important to inform public policy and health care planning. ARCOS tracks opioids from the point of manufacture to their point of sale or distribution at hospitals, practitioners, or retail pharmacies. State specific analyses may be underestimates if there were appreciable prescription opioids procured from out of state mail-order pharmacies. Although the increases in opioids for pain and an OUD exceeded the rapid rates of population elevations, ARCOS data is expressed in terms of overall drug weights. The body mass index of the population increased during the study period. One-quarter (26.3%) of Texas adults were obese in 2006 versus over one-third (33.0%) in 2017  (Warren, Beck & Rayburn, 2018). Although opioids are not typically dosed on a body weight basis  (Patanwala et al., 2012), obesity can contribute to lower back pain, osteoarthritis pain, and diabetic neuropathic pain. Other data sources (Alexander et al., 2019) will be needed to extend upon these results and identify changes in prescription opioid utilization among patients with specific indications. Future studies should evaluate whether state laws requiring checking the Texas PMP or a ten-day prescribing limit for acute pain achieves its desired objectives (Davis et al., in press; Collins et al., 2019; Texas House Bill 2174, 2019; Texas House Bill 3284, 2019; Sparks, 2019).

Conclusions

In conclusion, the Texas experience over the last decade with prescription opioids, particularly the dynamic changes with increases in buprenorphine and codeine but decreases in meperidine and hydrocodone, when coupled with population health indices (Forrester, 2012; Ng et al., 2017; Substance Abuse and Mental Health Services Administration, 2017; Piper et al., 2018; Andrilla, Coulthard & Larson, 2017; National Center for Health Statistics, 2017; Wright et al., 2014; Cabera et al., 2018), may be informative for public policy.

Supplemental Information

Supplemental Information 1 ARCOS opioid data by weight and population in Texas

Click here for additional data file.

Michael Sprintz, DO and Joseph Fraiman, MD provided feedback on earlier versions of this manuscript.

Additional Information and Declarations

Competing Interests

Author Contributions

Ethics

Data Availability

Brian J. Piper, Daniel Y. Chung and Stephanie D. Nichols were supported by the Fahs-Beck Fund for Research and Experimentation, a non-profit organization. Stephanie D. Nichols was a consultant for a research project supported by Shire. Brian J. Piper has a grant in review with Pfizer. The others authors have no relevant disclosures.

Ebuwa O. Ighodaro conceived and designed the experiments, performed the experiments, analyzed the data, authored or reviewed drafts of the paper, approved the final draft.

Kenneth L. McCall and Stephanie D. Nichols conceived and designed the experiments, authored or reviewed drafts of the paper, approved the final draft.

Daniel Y. Chung analyzed the data, contributed reagents/materials/analysis tools, prepared figures and/or tables, authored or reviewed drafts of the paper, approved the final draft.

Brian J. Piper conceived and designed the experiments, analyzed the data, prepared figures and/or tables, authored or reviewed drafts of the paper, approved the final draft.

The following information was supplied relating to ethical approvals (i.e., approving body and any reference numbers):

The University of New England deemed these procedures as exempt (#20180410-009).

The following information was supplied regarding data availability:

Raw data is available in the Supplemental File and at: https://www.deadiversion.usdoj.gov/arcos/retail_drug_summary/index.html.

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
