# Peer review of "Dynamic changes in prescription opioids from 2006 to 2017 in Texas"

_PeerJ, doi:10.7717/peerj.8108_

## Round 0.1 · original submission · Major Revisions

Both Reviewers consider the topic of this study worthwhile, although they differ in how they view the paper in its present form. For the study to considered further, please respond to each of the points made by both Reviewers. For the uninitiated (like me), it is probably worth explicitly discussing issues such as the potential differences between the ARCOS data and a "true" prescription monitoring program (and the conclusions that can be drawn). The data analysis is essentially repackaging of available data, and does not try and determine whether there are significant (i.e. statistically significant) differences in the ways that use of different opioids changes. Perhaps this is inappropriate, but if so, it would be good to explain why. The superficial comparisons with other states and the national picture is disappointing, and the final statement that the "Texas experience ... may offer some lessons for others to avoid or emulate" is quite ... wishy washy.

Reviewer 1 ·

Basic reporting

This article has potential, but it needs much work.
Footnote 1 cites a survey. I believe the correct survey should be the National Household Survey, Footnote 3 refers to areas with large military bases. I have never seen this article and wonder if there are not more robust research on the increases in population using pills. Suggest using local NSDUH or poison control data to be more meaningful. For footnote 5, you might find data showing that the number of methadone programs and population in treatment has been low, in comparison to NSSATS numbers. Footnote 6 refers to minorities. What proportion of heroin admissions were minorities and did NSDUH give any data on use by minorities. First paragraph should reflect the actual situation in Texas and not use data that does not seem relevant.
The second paragraph is incorrect. The State’s Prescription Monitoring Program was housed in the Department of Public Safety until September, 2017, when it was transferred to the State Board of Pharmacy. The Texas PDMP is now evolving a true PDMP. Suggest you reframe that paragraph to explain that the numbers until 2017 are describing an operation that really was not a PDMP.
The discussion about uninsured population is really not relevant. What about the numbers for states such as West Virginia? Be relevant.
Where did the numbers on methadone in Texas come from? I suspect the federal funding for opioid treatment was the driver of the increases in numbers, What is the source of the numbers in Figure 1B? Contrary to your statements, Texas has had a long history of not providing MAT. Provide numbers on how many methadone programs started because of the federal money, the increase in patient levels, how many physicians became qualified to prescribe buprenorphine, and the number of pharmacies that were dosing buprenorphine before and after the law changed.
Please describe how you accessed the ARCOS data. Did you have access to the ARCOS data set? If not, where did the numbers come from? How did you get the estimates for the various counties?Also, please make sure that statement cited n footnote 5 is a correct, given Texas did not have a true PDMP.
Do comment on the use of the federal funding to provide additional treatment. I seriously doubl that it would have happened without the federal initiative.
Figure 3 intrigues me. The counties that have historically had the highest use of hydrocodone and other opiates show decreases in Figure 3. I have followed some of those counties for years and been unable to fighre out the high use of opiates in them. Figure 4 raises questions because some of the areas with the highest “heat” already have active methadone program Other areas which did not have methadone now show high increases in buprenorphine. Can you cross-validate the increase in number of physicians now approved to prescribe buprenorphine with the increase in the drug? Use of NSSATS may help explain this finding.
This article takes a number of statements out of context with no relationship to the drug scene in Texas. It is pretty apparent to this reviewer that your knowledge of drug trends in Texas is limited. Your idea for a paper is interesting, but this paper is not relevant to the situation in Texa sand it did not provide me with an understanding of why some opioids decreased and others increased.

Experimental design

see above

Validity of the findings

see above

Additional comments

My comments on #1 include my concerns about the overall article.

·

Basic reporting

• In this article, opioid use in Texas is discussed which is of great importance as it is one of the States with the lowest opioid use and misuse. The introduction gives a good picture of opioid use in Texas however it would be interesting to consider people reading the article that may not well understand the American health system. For example, when discussing the lowest insurance rate (I guess we are talking health insurance here), it is important to highlight what it means in terms of Americans accessing the health system. In addition, a better link between sentences would benefit the introduction.
• Line 133: The sentence “Hydromorphone doubled while the increases in oxycodone was double the population” can be rewritten to increase clarity. I would suggest maybe add the number to the text to make it clearer, i.e.: Hydromorphone doubled from 2006 to 2017 while the increases in oxycodone of 46.6% was double the population change (+21.1%).
• Figure 1 – It is not clear to me why in the figure 1B oxymorphone and buprenorphine were not included.
• Figure 2 – I suggest mentioning that meperidine MME is too low to be visualised in the graph (it is colour-coded but not shown in the graph)
• Figure 3 and 4 – Is it possible to use different color schemes in these figures? It would be easier to visualise what the heat map means if decrease was shown in a cool color and the increase in red (as it is).
• Line 163: “…due to concerns about methadone and NTPs…” for clarity may be changed to “…due to safety concerns about the use of methadone in NTPs…”
• Line 165: “The rescheduling of hydrocodone…”, a further explanation of how was it rescheduled would help readers starting to work in the opioid field or not used to the scheduling system in the US.
• Line 176: The sentence “Prescription opioids were the first opioid…” should be rewritten for clarity and note that the ¾ claim is for 2000s data (it seems to have a shift since 2010). This data may be better presented discussing the shift from heroin to prescription opioids as the first opioid of abuse since the 1990s.
• Line 190: What does PMP (prescription monitoring program – add acronym to line 187) and CFR (Code of Federal Regulations?) stand for? 42 CFR part II is important to protect patient confidentiality. I am not sure that the way the sentence is written reflects the importance of the code. It sounds like it is disadvantaging the patient as they are “denied the benefit” of PMP, is it true? If so, can you further explain? Or is it disadvantaging data collection?
• In relation to Figure 3A, further discussing the difference in opioid use and misuse in between urban and rural areas would benefit the article. Keyes K et al 2014 discuss possible reasons why these differences may occur.
• Comment on how illegal opioid market, not accounted for in the database, can also affect the dynamic changes in prescription opioids. According to the 2018 National Drug Threat Assessment, Fentanyl and other synthetic opioids are primarily sourced from China and Mexico. According to figure 2B, Fentanyl decreased in the period researched. Could it be a reflection of availability from other sources?
• You may want to include in the discussion comments on the recent Law approved in Texas restricting the opioid length of supply for acute pain and how do you think this fit with the data presented.
• Reference 24: add access date

Experimental design

This article presents an original primary research within the Scope of the Journal. The aim of this research is clear and meaningful due to the opioid crisis in the US. It is crucial to research States where the opioid crisis isn’t as bad to support policies and interventions on the more affected States.

Validity of the findings

The database used is comprehensive although limitations exist, and they were raised in the discussion session. If the data base is publicly accessible, a reference to it with a link should be added to the reference list.
Aims were met and succinctly mentioned in the concluding session. However, the first part of the conclusion is not the conclusion of this study, therefore it needs to be rewritten to clarify the conclusion of this study compared to statements based on previous studies or assumptions.

Additional comments

I commend the authors for this work. The manuscript is well written and mostly easy to understand. The main weakness is at some points it feels like the information was not joined together.

---

## Round 0.2 · accepted · Accept

I apologise for the delay in this decision, I have been travelling for work. Thank you for your efforts in constructively addressing the comments of the Reviewers.